# Targeting Myeloperoxidase Activity and Neutrophil ROS Production to Modulate Redox Process: Effect of Ellagic Acid and Analogues

**DOI:** 10.3390/molecules28114516

**Published:** 2023-06-02

**Authors:** Gilles Degotte, Michel Frederich, Pierre Francotte, Thierry Franck, Thomas Colson, Didier Serteyn, Ange Mouithys-Mickalad

**Affiliations:** 1Laboratory of Medicinal Chemistry, Center of Interdisciplinary Research on Medicines (CIRM), University of Liege, 4000 Liège, Belgium; dgilles@hotmail.be (G.D.); pierre.francotte@uliege.be (P.F.); t.colson@uliege.be (T.C.); 2Laboratory of Pharmacognosy, Center of Interdisciplinary Research on Medicines (CIRM), University of Liège, 4000 Liège, Belgium; m.frederich@uliege.be; 3Centre for Oxygen, Research & Development (CORD), Center of Interdisciplinary Research on Medicines (CIRM), University of Liège, 4000 Liège, Belgium; t.franck@uliege.be (T.F.); didier.serteyn@uliege.be (D.S.); 4Department of Clinical Veterinary, Equine Clinic, Large Animal Surgery, Faculty of Veterinary Medicine, University of Liège, 4000 Liège, Belgium

**Keywords:** malaria, ellagic acid, polyphenols, reactive oxygen species, PMNs, horseradish peroxidase, myeloperoxidase

## Abstract

Malaria is an infectious disease caused by a *Plasmodium* genus parasite that remains the most widespread parasitosis. The spread of *Plasmodium* clones that are increasingly resistant to antimalarial molecules is a serious public health problem for underdeveloped countries. Therefore, the search for new therapeutic approaches is necessary. For example, one strategy could consist of studying the redox process involved in the development of the parasite. Regarding potential drug candidates, ellagic acid is widely studied due to its antioxidant and parasite-inhibiting properties. However, its low oral bioavailability remains a concern and has led to pharmacomodulation and the synthesis of new polyphenolic compounds to improve antimalarial activity. This work aimed at investigating the modulatory effect of ellagic acid and its analogues on the redox activity of neutrophils and myeloperoxidase involved in malaria. Overall, the compounds show an inhibitory effect on free radicals as well as on the enzyme horseradish peroxidase- and myeloperoxidase (HRP/MPO)-catalyzed oxidation of substrates (L-012 and Amplex Red). Similar results are obtained with reactive oxygen species (ROS) produced by phorbol 12-mystate acetate (PMA)-activated neutrophils. The efficiency of ellagic acid analogues will be discussed in terms of structure–activity relationships.

## 1. Introduction

Ellagic acid (2,3,7,8-tetrahydroxy-chromeno[5,4,3-cde] chromene-5,10-dione, Figure 1, see below) is a very common polyphenol belonging to the class of shikimates. It is a precursor to or byproduct of ellagitannins, found in abundance in pomegranate (*Punica granatum*) [1]. Among other biological actions, it is a very effective inhibitor of *Plasmodium falciparum*, showing an in vitro activity ranging from 100 to 300 nM [2]. Ellagic acid also has curative activity in vivo after intraperitoneal injection [3]. In previous work carried out in the CIRM Pharmacognosy Laboratory (ULiège, Belgium), we have also shown that different derivatives of ellagic acid obtained from a type of punicalagins isolated from a Rwandan plant (*Terminalia mollis*) displayed antimalarial activity in vivo [4]. Ellagic acid is also a powerful antioxidant, contributing to the potent activities of many fruits and vegetables, as well as to their presumed antimutagenic properties [5,6,7]. Likewise, Mohanty’s group has shown the protective effect of ellagic acid in LPS-induced inflammatory cytokine storms and oxidative stress induced by malaria disease [8].

Malaria is one of the leading infectious causes of death worldwide, particularly in sub-Saharan Africa. It is also one of the deadliest infectious diseases caused by parasites of the *Plasmodium* genus. Annually, more than 247 million clinical cases are reported worldwide, and there were 627,000 deaths from malaria in 2021 [9,10,11]. Among them, pregnant women and children under the age of 5 are the most vulnerable groups affected by malaria [12]. The disease usually occurs in the tropical regions of Africa, Southeast Asia, and Latin America, although around 90% of malaria deaths are recorded in Africa. Malaria is defined as endemic in several countries where around 2.4 billion people live, representing 40% of the world’s population [13]. Malaria-related deaths reduced steadily over the period from 2000–2019, from 897,000 in 2000 to 577,000 in 2015, and to 568,000 in 2019. In 2020, the mortality increased by 10% compared to 2019, to an estimated 625,000, and to 619,000 in 2021. An increase in resistance to most antimalarials, especially the artemisinin derivatives, and the existence of adverse side effects are prompting researchers to find new therapeutic alternatives [14].

One of the potential targets for new drugs could be the redox process involved in the parasite life cycle. Indeed, during the development of the parasite and the infection’s phase of red blood cells, a redox cycle involving heme and methemoglobin (metHb) takes place and promotes the spread of the parasite, which is then followed by an inflammatory phase in which the immune system of the host is compromised [15]. Among the cells involved in this immune defense, neutrophils play an important role. Produced by the bone marrow, they are, in fact, the most abundant leukocytes in the blood. Their major role is to defend the organism by killing pathogens through the mechanism of phagocytosis [16].

During phagocytosis, several processes occur, including: (i) the production of reactive oxygen species (ROS) through NADPH oxidase activation [17]; (ii) neutrophil degranulation, causing the release of enzymes such as myeloperoxidase (MPO) and elastase; and finally, (iii) neutrophil extracellular trap-osis (NETosis), a process that allows for the formation of neutrophil extracellular traps (NETs) [18,19,20].

Consequently, neutrophils play a key role in malaria, since parasites invade red blood cells and trigger the systemic activation of neutrophils [21,22]. The formation of NETs is the direct consequence of the activation of polymorphonuclear leukocytes (PMNs) when the organism is infected. Neutrophils migrate from the bloodstream to the site of infection as the first line of defense against the pathogen. The neutrophils’ arsenal, which makes it possible to carry out this defense, consists of a significant number of antimicrobial substances in their granules. These two phenomena are involved in NETosis and seem to have a beneficial role in confining the parasite and slowing its spread. However, during the acute onset of malaria, this process can be exacerbated, causing harmful effects. Therefore, the inhibition of NETs could be considered as a treatment for the vascular events occurring during severe malaria [19]. Finally, considering these issues, and given the involvement of enzymes such as MPO and ROS in the formation of NETs, the inhibition or modulation of the action of PMNs could be an interesting therapeutic approach.

A recent study by Muchtar et al. [23] reported one of the mechanisms of action of ellagic acid as acting through the pH of the *P. falciparum* digestive vacuole. Our group recently reported that polyphenolic derivatives, analogues of ellagic acid, constitute a promising scaffold that could play crucial role in malaria treatment through a currently unknown mode of action [24].

Ellagic acid acts preferentially at the trophozoite and young schizont stages of the parasite, when hemoglobin digestion takes place as well as an abundant synthesis of proteins and nucleic acids. This can clearly be altered by oxidative stress, leading to the death of the parasite. It is well recognized that many polyphenols display both anti- and pro-oxidative properties depending on the chemical environment and concentration [25]. Clinically, it has also been demonstrated that the addition of antioxidants to conventional antimalarial therapy has beneficial effects because of this dual activity [26]. Consistent with this, ellagic acid was recognized as impairing several antioxidant mechanisms, including the glutathione cascade.

The main issue with ellagic acid remains its very low oral bioavailability (<30% in mice), which impedes its use as an oral antimalarial drug and was partially linked to its low hydrosolubility. To overcome this problem, the pharmacomodulation of this scaffold and the synthesis of new polyphenolic compounds were undertaken to improve the *per os* antimalarial activity. The present work aimed at investigating the modulating effect of ellagic acid and its analogues on the redox activity of neutrophils and myeloperoxidase involved in malaria onsets. Therefore, different models were used to first screen the anti-radical property of ellagic acid and its analogues, and then to investigate the antioxidant activity using both enzyme-catalyzed oxidation models and the ROS produced by stimulated neutrophils.

## 2. Results and Discussion

### 2.1. 2,2′-Diphenyl-1-picrylhydrazyl (DPPH) Assay

Appendix A shows that most of the tested molecules display quite a similar inhibiting effect in the DPPH test, explaining their ability to reduce the DPPH radical by more than 60 to 70% at a final concentration between 5 µM and 10 µM, whereas at the lowest concentration of 1 and 2 µM, the inhibition percentage comprises between 20 to 48% (Appendix A). This strong activity seems to be linked to the presence of phenolic functions, which must therefore remain free, as shown by the total loss of anti-radical effect for compound **6** with very low inhibiting activity (3 to 15%), reaching even −3% at the highest concentration. In this latter case, the phenols of ellagic acid have been converted into a carbamate function. However, it is noted that there is no significant impact of dimerization on this activity, since protected methyl gallate has a similar potency as 3,4,5-trihydroxybenzoic acid dimers such as **10** or **12**, as shown in Appendix A. Moreover, it seems that neither the nature of the linker nor the total number of aromatic hydroxyls have any influence on the activity. The IC_50_ values of the DPPH assay are listed in Table 1.

### 2.2. 2,2′-Azino-bis(3-ethylbenzothiazoline)-6-sulfonic Acid (ABTS) Assay

As with the DPPH assay, most of the tested compounds and both EA references exhibited higher and dose-dependent inhibiting activity (Appendix A). During the ABTS test, another range of concentrations (10^−8^ M, 10^−7^ M, 10^−6^ M, and 10^−5^ M) was used, as we observed a strong inhibiting activity with the previous concentrations used for the DPPH assay. It is noted that the tested compounds showed a higher activity than in the DPPH test, mainly at the highest concentration of 10^−5^ M, reaching almost 80 or 85% inhibition. However, at the two lowest concentrations of 0.01 and 0.1 µM, the inhibition was weak. Although the IC_50_ values for the two assays are roughly quite similar, still, compounds **2**, **4**, **7**, and **12** exhibited the best inhibition for the DPPH assay, with IC_50_ values of 2.86 ± 0.13 µM, 3.36 ± 0.25 µM, 3.14 ± 0.72 µM, and 3.70 ± 0.45 µM, respectively. The IC_50_ values for the ABTS test for the same compounds were lower: 5.50 ± 0.16 µM, 5.77 ± 0.01 µM, 4.68 ± 0.13 µM, and 5.86 ± 0.18 µM, respectively (Table 1).

Again, this effect can certainly be linked to the presence of an unprotected catechol or pyrogallol ring, since the only compound with no significant activity is the prodrug carbamate derivative of ellagic acid (**6**). However, although the results obtained are very similar to those of the DPPH test, it can be noticed that the total number of phenol functions seems to also have importance in terms of gallic acid dimers’ activity. However, the derivative **13**, with a naked benzoic acid nucleus, still exhibited a better effect (IC_50_ between 4.91 ± 0.26 µM for DPPH and 6.0 ± 0.31 µM for ABTS) than its substituted homologue **3**, but it was less active than **2** (IC_50_ of 2.86 ± 0.13 µM for DPPH 5.52 and ± 0.34 µM for ABTS). In contrast, the nature of the linker used for the dimerization has little impact on the antioxidant activity. The IC_50_ values of the DPPH and ABTS tests are listed in Table 1.

### 2.3. Peroxidase Activity

#### 2.3.1. Horseradish Peroxidase (HRP) Test

The enzyme-catalyzed oxidation of the substrate (L012) in the presence of H_2_O_2_ results in an enhanced light emission. Regarding the inhibition of oxidation of the L012 probe, mediated by horseradish peroxidase, we found similarities with the previous tests (DPPH and ABTS). Indeed, the presence of substituted phenols is again essential given the lack of effect for compound **6** on the electron exchange during the peroxidase activity (Appendix A). During this assay, the final concentrations used were the ones for the DPPH test. Indeed, when the concentration range used for ABTS test was applied, we found that the peroxidase enzyme was functioning too fast, and the tested samples were not inhibitory enough.

The two standards of ellagic acid **1a** (Sigma, St-Louis, MO, USA) and **1b** (Lab) exhibited a strong inhibiting activity, and most of the polyphenols had a similar dose-dependent inhibiting effect (Appendix A). Surprisingly, one can note that compound **7** displays a reduced antioxidant effect when compared to its congeners (**2**, **3**, **4**, and **8**, bearing two, three, and four carbon-linkers, respectively). Likewise, the dimer compound **12**, with two amide functions and a three-carbon spacer, also showed very good inhibition (Appendix A). The effect of protected methyl gallate (**11**) in this experiment is quite like some dimer compounds of polyhydroxybenzoic acids. Altogether, our results with these first three models indicate that the essential structural requirement for the activity remains the presence of free phenol groups, whereas the nature of the carbon linkers, including amide or ester function, does not have a significant impact on this inhibiting activity.

The IC_50_ values found in the three different models are not too distant from each other, indicating that most of the compounds display good peroxidase-inhibiting activity and free radical scavenging, except for compound **6**.

Based on the HRP results, it was interesting to investigate the effect of ellagic acid and its polyphenol analogues on another peroxidase enzyme of interest, myeloperoxidase (MPO). Indeed, the peroxidase activity differs from one enzyme to another, and the native structure, as well as the type of the active site, could influence the reactivity with the substrate and potential inhibitors.

#### 2.3.2. Myeloperoxidase (MPO) Activity (Classical Assay)

Myeloperoxidase works in a similar way as HRP, but in addition to peroxidase activity, MPO drives the chlorination reaction in the presence of a chlorine atom (Cl). As buying MPO is quite expensive, the amount employed for this test is generally low. Therefore, we selected the same concentration range as that for the ABTS assay (0.01 to 10 µM). The MPO classical peroxidase assay uses hydrogen peroxide as a substrate and Amplex Red (AR) as a fluorescence probe, instead of the L012 used for HRP.

Regarding the inhibition of human MPO activity, we again noticed the lack of effect of **6**, which can surely be linked to the absence of free phenols on the tetracyclic structure. However, in contrast to the HRP assay, here, compound **7** displays a nice dose-dependent inhibiting effect, quite like those observed for **2**, **3**, **4**, and **5**, with inhibition values reaching 94%. For the second series of polyphenols (see Figure 1B below), including compound **13**, which has a naked benzoic acid pattern, their inhibition activity at low concentrations of 0.01 and 0.1 µM was much lower than that seen for the first series (see Figure 1A below). Similar to the first assays, the linker does not influence the peroxidase activity.

However, contrary to what may have been seen in previous experiments, the position of phenols on the aromatic core would not impact the activity, irrespective of the number of carbon atoms of the linker. Thus, compound **3**, with only one pyrogallol/catechol nucleus, displays more pronounced inhibiting activity (98%) compared to compound **13**, which does not have hydroxyl groups on the second aromatic ring ranging from 40 to 84%, with IC_50_ values of 3.19 ± 0.07 µM for **3** and 5.46 ± 0.01 µM for **13**. Finally, it should be noted that in the case of this experiment, the protected methyl gallate derivative (**11**) displays the same activity profile as other compounds at a higher concentration (1 and 10 µM), but it is weak at low concentrations (0.01 and 0.1 µM). Given its potential for electronic delocalization, because of these four aromatic cycles, ellagic acid is still a powerful molecule, but with a less pronounced effect in the present model than the dimers **2**, **3**, **4**, **5**, **7**, **10**, and even **8**.

Overall, except for the low inhibition effect seen with **6**, which can be explained by the lack of free phenols in its structure, most of the tested compounds behaved as relatively good electron donors in the peroxidase cycle, hindering the turnover of the redox cycle. Some of them, such as compound **2**, **4**, **5**, **7**, **8**, and **10** even displayed high potency, more so than ellagic acid, at lower concentrations.

#### 2.3.3. MPO Activity by Specific Immunological Extraction Followed by Enzymatic Detection (Siefed Assay)

Based on the results obtained with the peroxidase (classical assay), it appears that **1** and most of the tested compounds interact with either compound I, a π-cation radical of porphyrin, or with the product of oxidation of Amplex Red into resorufin. This reaction is performed in situ and does not consider the enzyme inhibitor contribution. To investigate the direct interaction of ellagic acid or polyphenol analogues with the enzyme, another experiment was designed based on the use of a specific ELISA, called SIEFED, which was previously developed in our laboratory [27]. Similar to that which we observed with the previous test, the structural trends were quite well-correlated. The weak activity or the absence of inhibition activity for **6** remains quite similar to the previous assays. Surprisingly, in this experiment, the protected methyl gallate (**11**) behaves as an activator of MPO activity by increasing the fluorescence emission (see Figure 2B below). Similarly, it seems that the presence of many free phenols is a requirement for peroxidase-inhibiting activity, since we observed a reduced effect of compound **13**, bearing on one side a simple functionalized benzoic acid, in comparison with the equivalent compound **2**. However, the dimerization of polyhydroxybenzoic acids again demonstrates the acid’s interest compared to gallic acid, since the majority of open ellagic acid derivatives are more effective. Thus, compound **3**, with only one pyrogallol/catechol nucleus, displays more pronounced inhibiting activity compared to compound **13**, which does not have hydroxyl groups on the second aromatic ring. This is reminiscent of the metal chelation property of phenol or polyphenols having catechol groups in their structures. Indeed, these polyphenol analogues are able to react with ROS by two mechanisms, including electron transfer (ET) and hydrogen atom transfer, (HAT) but also by chelating the metal ions (e.g., Fe(II)/Fe(III)) that are released during the metabolism dysfunction. These compounds may also interact with enzymes in the active site if they can reach the canal and establish a link between amino acids’ residues that form the porphyrin, as observed in the SIEFED assay.

Finally, it should be noted that in the case of this experiment, the protected methyl gallate derivative (**11**) displays an activity profile quite contrary to other compounds, since it seems to have no activity or even to activate the oxidation of the probe by human myeloperoxidase. Consequently, in certain cases, the orientation of the molecule in the enzyme active site may explain the unexpected effect that was observed with the protected methyl gallate derivative.

It can also be noted that among dimers, compounds with an aromatic linker, such as **7** and **10**, appear to be significantly more powerful than their linear equivalents, except for **3** and **4**, since their MPO activity was inhibited in quite the same way. In this model, these two compounds (**7** and **10**) are even superior to ellagic acid. It can be assumed that additional π-stacking interactions improve the affinity for the enzyme. Moreover, it seems that the nature of the spacer (carbon or aromatic), as well as the phenolic nuclei, has less importance in terms of activity power towards MPO (for the protected compounds **11** and **13** in the SIEFED assay).

It is well known that neutrophils play a key role in the redox process during the development of malaria. Moreover, ellagic acid has been described for its anti-radical and antioxidant properties like other polyphenols. Therefore, the behavior of ellagic acid and its polyphenol analogues on the human enzyme (MPO) model indicates that the fate of those molecules in vivo may be different. These interesting results prompted us to test their potential effects on the ROS produced by stimulated neutrophils, mimicking the in vivo situation.

### 2.4. Cellular Antioxidant Activity

Like previous assays, most of the tested compounds exhibited an inhibiting activity in a dose-dependent manner, irrespective of the number of free phenols and the nature of the linker (see Figure 3 below). It is worth noting that **6** compound was again less active than all the other tested compounds, which tends to confirm the need for free phenol functions to be able to interact with the target (e.g., ROS). Herein, the inhibition effect with the highest concentration of **6** reached 58%, which can be attributed to an action on the cell membrane through an effect on the NADPH oxidase enzyme. Interestingly, in this cellular model, protected methyl gallate (**11**) showed a good inhibitory effect similar to most of the tested dimers. Consequently, there seems to be no advantage to using polyhydroxybenzoic acid dimers. Thus, neither the number nor the position of the aromatic hydroxyls seem to have any significant importance here; the same is true for the linker. On the other hand, de-planarized derivatives are sometimes more powerful than the tetracyclic molecule when the effect at the intermediate concentration of 0.1 µM is compared. The polyphenols are well-known for their anti-radical and antioxidant properties. Depending on the redox potential in the studied reaction medium, their activity can be changed, since at low concentrations, they can act as pro-oxidants [28]. Ellagic acid is widely studied and has shown a potential therapeutic interest in the management of malaria [8]. The mechanism of action of ellagic acid is not clearly established, but it is known that the molecule, when in contact with the parasite environment, could become pro-oxidant and efficient. Still, its low solubility remains a problem that needs to be solved by either the pharmacomodulation of the ellagic acid structure, or by preparing a water-soluble form by means of cyclodextrin or nanoparticles [29,30]. Herein, structural analogues of ellagic acid were designed and synthetized to overcome the problem of poor water solubility as well as the limited bioavailability that hampers clinical applications. Most of the dimers studied have shown a dose-dependent inhibitory activity irrespective of the linker. Having flexible dimers seems to be beneficial, as in general, most of the compounds are more soluble than ellagic acid, but we found that the aromatic substituents negatively impact solubility. The spacer constitutes the parameter that can define the solubility. These findings also indicate that the presence of free phenols in the compounds’ structure is very important, even compulsory, to allow for electron or hydrogen atom transfer and to reduce the oxidative stress. This is the case with compound **6**, which has no free phenols in its structure. However, the discrepancy observed in the SIEFED assay can be explained by the fact that in this model, the compound is first incubated with the enzyme (MPO) and then removed, if not bound to the enzyme, by the washing step that investigates whether the interaction between the complex MPO compound will drive the peroxidase activity. Indeed, some of the compounds with different planar conformations promote, rather than inhibit, MPO activity, such as compound **13** and the protected compound **11** (see Figure 3 below). The general structure–activity relationship (SAR) findings can be discussed as follows:

Regarding the antioxidant activity of ellagic acid and some derivatives, all the experiments carried out in this study have led to solid conclusions as to the essential structural elements. On the one hand, there is a need for non-replaced phenol functions to maintain a significant effect, as shown by the overall lack of activity of sevellagic acid (**6**). On the other hand, it can also be concluded that it is useful to use gallic acid dimers, since in most experiments, these have shown an effect that is superior or equal to **11**. This holds true despite the loss of aromatic hydroxyls in some cases. However, it is preferable to maintain catechol or pyrogallol nuclei. Indeed, products with a benzoic acid or 3,5-dihydroxybenzoic motif are sometimes significantly less powerful in some experiments.

## 3. Conclusions

Finally, it is interesting to note that, similar to that which was deduced for antiplasmodial activity [24], the linker used has little impact on the overall power of the different compounds. As for future perspectives, a docking study may help to determine the interaction with the enzyme and to better explain the inhibition or pro-oxidant activity of certain compounds. Furthermore, in view of these results, a further specific study on the formation of NET, which is very involved in certain cases of severe malaria, is to be considered. The results obtained on enzyme and cellular models can be extrapolated to NET formation, because both neutrophils and MPO play a crucial role during this formation.

## 4. Materials and Methods

### 4.1. Reagents

All the salts used to prepare the buffered solutions, methanol and ethanol were of analytical grade from Merck VWR (Leuven, Belgium). The chemiluminescent probe, L0-12 (8-amino-5-chloro-7-phenyl-pyrido[3,4-d] pyridazine-1,4(2H,3H)dione), was obtained from Wako Chemical Europe (Neuss, Germany); 2,2′-Diphenyl-1-picrylhydrazyl (DPPH), 2,2′-azino-bis(3-ethylbenzothiazoline)-6-sulfonic acid (ABTS), Amplex Red, sodium nitrite (NaNO_2_), sodium persulfate (Na_2_S_2_O_8_), and hydrogen peroxide (H_2_O_2_) were all purchased from Sigma-Aldrich (Steinheim, Germany). Phorbol 12-myristate 13-acetate (PMA) was purchased from Sigma (St. Louis, MO, USA). Horseradish peroxidase (HRP) and Tween-20 were from Merck (Darmstadt, Germany), bovine serum albumin (BSA) was from Roche Diagnostics Gmbh (Mannheim, Germany), and human myeloperoxidase (MPO) was purchased from Calbiochem Millipore (Bellirica, Madison, WI, USA). The two ellagic acids used as standards were from Sigma-Aldrich (Darmstadt, Germany) and the Laboratory of Medicinal Chemistry (ULiège, Belgium), respectively. All the polyphenol compounds, analogues of EA, were synthetized by the Laboratory of Medicinal Chemistry (ULiège, Belgium).

Ellagic acid (2,3,7,8-tetrahydroxy-chromeno[5,4,3-cde] chromene-5,10-dione) was purchased from Sigma (**1a**) and synthetized by the LPC ULiège–Belgium (**1b**). Most of the polyphenolic analogues were provided by the LPC, and their synthesis has already been published [24]. The procedure used to obtain compound **11** is described in the Appendix A of the present article. The following chemical names are numbered and listed below: (**1**) 2,3,7,8-Tetrahydroxy [1]benzopyrano [5,4,3-cde][1]benzopyran-5,10-dione; (**2**) propane-1,3-diylbis(3,4,5-trihydroxybenzoate; (**3**) 3-((3,5-dihydroxybenzoyl)oxy)propyl 3,4,5-trihydroxybenzoate; (**4**) ethane-1,2-diyl bis(3,4,5-trihydroxybenzoate; (**5**) 3-((3,4-dihydroxybenzoyl)oxy)propyl 3,4,5-trihydroxybenzoate; (**6**) Sevellagic acid; (**7**) 4-(3,4,5-trihydroxybenzamido)phenyl 3,4,5-trihydroxybenzoate; (**8**) butane-1,4-diyl bis(3,4,5-trihydroxybenzoate; (**9**) 3-(3,4,5-trihydroxybenzamido) propyl 3,4,5-trihydroxybenzoate; (**10**) 3-(3,4,5-trihydroxybenzamido)phenyl 3,4,5-trihydroxybenzoate; (**11**) methyl 2-ethoxy-7-hydroxybenzo[d][1,3]dioxole-5-carboxylate; (**12**) *N*,*N*′-(propane-1,3-diyl) bis(3,4,5-trihydroxybenzamide; (**13**) 3-(benzoyloxy)propyl 3,4,5-trihydroxybenzoate.

### 4.2. In Vitro Radical Assays

#### 4.2.1. ABTS Test

The ABTS assay was based on the method described by Re et al. [31]. For the generation of ABTS^•+^ radicals, sodium persulfate (2.45 mM) aqueous solution was mixed with ABTS (7 mM) and incubated overnight in the dark to obtain a dark-colored solution. Stock solution of ABTS^•+^ was then diluted by adding pure methanol (100%) to obtain an absorbance of 0.70 (±0.02) at 734 nm at 30 °C. To obtain more repetition, the ABTS assay was performed using a microplate reader (ThermoLabsystem, Finland). An aliquot of 2 µL of the tested compound (EA or analogue) was added to 198 µL of ABTS^•+^, and the decrease in absorbance was monitored at 740 nm after 30 min [32]. During this reaction, the blue-green ABTS radical cation is converted back into its colorless neutral form in the presence of the potential antioxidant molecule. A control consisted of 2 µL of DMSO in 198 µL of ABTS^•+^ solution. The reducing capacity was determined according to the formula:% inhibition = (A_control_ − A_Sample_) × 100/A_Control_(1)

#### 4.2.2. DPPH Test

The DPPH assay was performed according to the method developed by Brand-Williams et al. [33], though slightly modified [32]. A solution of 1 mM DPPH in absolute methanol was stirred for 40 min. Absorbance of the solution was adjusted to 0.650 ± 0.020 at 517 nm using fresh absolute methanol. For replicates, the DPPH assay was performed using a microplate reader. For this, 2 µL of EA taken as a standard or sample (EA analogue) were mixed with 198 µL of DPPH solution and incubated for 30 min in the dark covered with aluminum foil. When reacting with an antioxidant, the DPPH• radical is converted into DPPH, and its color changes from purple to yellow. The antioxidant effect may be easily evaluated by observing the decrease in visible absorption. The absorbance decrease was monitored at 510 nm for 30 min with a Multiskan Ascent 96-plate reader (ThermoLabsystem, Finland). A control consisted of 2 µL of DMSO in 198 µL of DPPH solution. A similar formula (Equation (1)) was applied to determine the DPPH radical-scavenging activity.

### 4.3. Peroxidase Activity

#### 4.3.1. HRP Assay

Horseradish peroxide (HRP) reacts with hydrogen peroxide (H_2_O_2_), yielding “compound I (HRP^+**.**^-Fe (IV=O))”, which is a reactive intermediate of HRP that is able to oxidize any potential reducing agent. Briefly, 2 μL of compound or reference molecule (ellagic acid) were incubated in phosphate buffer at pH 7.4 (153 μL, 50 mM) containing 5 μL HRP (100 µg/L), 20 μL L0-12 (10^−3^ M), and 20 μL H_2_O_2_ (1 mM), and the chemiluminescence was immediately measured using a Fluoroskan Ascent 96-plate reader (ThermoLabsystem, Finland) at 37 °C for 30 min [32,34]. The analysis was conducted in a 96-well microtiter plate. All the assays were performed in triplicate and repeated at least twice (N = 2; n = 6). The relative activity in terms of inhibition percentage of the tested compound or reference (ellagic acid) was compared to a control in the presence of DMSO taken as a solvent. The inhibition percentage was calculated using a similar formula (Equation (1)) as described above.

#### 4.3.2. MPO Assay

Measurement of myeloperoxidase activity

Measurement of the peroxidase activity of MPO was performed by a classical enzymatic assay and by SIEFED assay as described by Nyssen et al. [35]. The MPO solution was prepared with purified human MPO in dilution buffer (PBS 20 mM at pH 7.4 with 5 g/L BSA and 0.1% Tween-20). The solutions of EA or analogue, at final concentrations ranging from 10^−8^ M to 10^−5^ M, were incubated for 10 min with human MPO at a final concentration of 5 mU/mL before further use. The revelation of MPO activity was performed by monitoring the enzyme-catalyzed oxidation of Amplex Red in the presence of H_2_O_2_ (10 μM) and nitrite (4.5 mM) in phosphate buffer at pH 7.4.

Classical assay of MPO activity

After incubation, mixtures containing 100 μL of EA/analogue or vehicle (DMSO) and MPO were loaded into wells of microtiter plates (transparent), and the peroxidase activity was measured by adding 10 μL sodium nitrite solution (4.5 mM, final concentration) and 100 μL of the reactional solution containing 10 µM H_2_O_2_ and 40 µM Amplex^®^ Red (AR) in phosphate buffer (50 mM) at pH 7.4. The oxidation of AR into the fluorescent resorufin adduct (λexcitation = 544 nm; λemission = 590 nm) was monitored for 30 min at 37 °C with a fluorescent plate reader (Fluoroskan Ascent, Fisher Scientific, Hampton, NH, USA). To eliminate the possibility of the artifact reactions that might arise from MPO activity or its natural substrate (H_2_O_2_), the direct reaction of ellagic acid or an analogue with H_2_O_2_ was performed in phosphate buffer (PBS) without the addition of human MPO.

SIEFED assay of MPO activity

Samples with MPO and various concentrations of juglone were prepared and incubated as with the classical assay. An amount of 100 μL of each mixture (MPO alone or MPO + EA/DMSO) was then loaded into the wells of a SIEFED microtiter plate coated with rabbit polyclonal antibodies (3 μg/mL) against human MPO and incubated for 2 h at 37 °C in darkness. After washing up the wells, the activity of the enzyme captured by the antibodies was measured by adding 10 μL sodium nitrite solution (4.5 mM, final concentration) and 100 μL of a reactional solution containing 10 μM H_2_O_2_ and 40 μM Amplex^®^ in phosphate buffer (50 mM) at pH 7.4. The oxidation of Amplex^®^ Red into the fluorescent adduct resorufin (λexcitation = 544 nm; λemission = 590 nm) was monitored for 30 min at 37 °C with a fluorescent plate reader (Fluoroskan Ascent, Fisher Scientific). As for the MPO direct assay, a control assay set as the relative value of MPO activity was performed with purified MPO in the presence of PBS instead of the samples of EA or analogue dissolved in DMSO. In this SIEFED assay, MPO was bonded by the antibodies into the wells, and EA/analogue was discarded in the washing step before starting the measurement of the enzymatic activity. For both MPO assays, the inhibition percentage was calculated using a similar formula (Equation (1)) as described above.

### 4.4. Cellular Assays

Equine neutrophils were isolated from whole blood using EDTA disodium salt (1.6 mg/mL) as an anticoagulant. The blood was drawn from the jugular vein of healthy horses bred and fed under identical conditions without medical treatment. All the experiments were realized with the approval of the ethics committee of the Faculty of Veterinary Medicine of the University of Liege (agreement number 1474). Briefly, the neutrophils were isolated at room temperature (18–22 °C) by centrifugation (400× *g*; 45 min; 20 °C) on a discontinuous Percoll density gradient according to the method previously described [35]. The cells were gently collected and washed with two volumes of physiological saline solution. After the supernatant removal, the cell pellets were re-suspended in 2 mL PBS and counted for further use.

#### Neutrophil ROS Production by Chemiluminescence Investigation

The ROS produced by PMA-activated neutrophils were measured by L012-enhanced chemiluminescence (CL) under an adaptation of the method previously described by Benbarek et al. and Derochette et al. [36,37]. Neutrophil suspensions were distributed in the wells (neutrophils/well, 143 μL PBS) of a 96-well microtiter plate (White Combiplate 8, Fisher Scientific) and incubated for 10 min at 37 °C with 2 μL of EA or analogue to reach final concentrations of 10^−8^ M, 10^−7^ M, 10^−6^ M, and 10^−5^ M. After incubation, 25 μL CaCl_2_ (10 mM) and 20 μL L-012 (10^−4^ M) were added into the wells (final volume of 200 μL). Then, cell suspensions were activated with 10 μL PMA (16 μM) just before CL measurement [38]. The CL response of the neutrophils was monitored for 30 min at 37 °C with a Fluoroskan Ascent spectrophotometer (Fisher Scientific, Tournai, Belgium) and expressed as the integral value of the total CL emission. A control was performed with neutrophils activated with PMA (positive Ctrl) in the presence of PBS instead of ellagic acid analogue. Another control was performed with PMA-activated neutrophils in the presence of the vehicle solution of ellagic acid or analogue (1% DMSO) (DMSO, final concentration) and was taken as 100% CL response. The negative control was performed with unstimulated neutrophils (NA) with the CL probe alone. The inhibition percentage was calculated using a similar formula (Equation (1)) as described above.

## Data Availability

Not applicable.

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
