# Peer review of "Targeting Myeloperoxidase Activity and Neutrophil ROS Production to Modulate Redox Process: Effect of Ellagic Acid and Analogues"

_molecules, 2023, doi:10.3390/molecules28114516_

Round 1

Reviewer 1 Report

The article entitled "Targeting Myeloperoxidase activity and Neutrophil ROS production to tackle Malaria Disease: Effect of Ellagic acid and analogues" by Degotte Gilles, Michel Frederich, Pierre Francotte, Thierry Franck, Thomas Colson, Didier Serteyn and Ange Mouithys-Mickalad describes the study of the redox properties of a large series of analogues derived from ellagic acid. More precisely, the authors examined the inhibitory effects on free radicals (antioxidant activities with DPPH and ABTS) as well as their impact on the oxidation of substrates catalysed by peroxidase-type enzymes (HRP/MPO). The authors highlighted the importance of the presence of free phenols and the weaker impact of the spacer used through structure-activity relationships. This article is essentially a continuation of a recently published work with a detailed description of the rational design, synthesis, in vitro and in vivo antimalarial activities and in vitro and in vivo cytotoxicity. The authors never proposed a possible activity involving neutrophils, ROS and peroxidases in this accepted MS. How and why did the authors reach this possibility? While ellagic acid has an antimalarial activity in vitro of 100-300 nM, the synthesised derivatives are only effective in the range of several µM.

- Introduction. Chloroquine is no longer used except against P. vivax only where it is still susceptible to this drug. It is rather the emergence of resistance against artemisinin derivatives that is of great concern.

- Have the authors made any progress in studying the mechanism of action (MoA)? They suggest that these compounds may act in redox processes but what evidence is there to support their suggestions. Ellagic acid could act by other MoAs (e.g. haemozoin inhibition).

- The authors never discuss the ability of these systems to complex metal cations such as iron(III)/iron(II) or possibly (met)haemoglobin. Antioxidant action can also come from the ability of a ligand to complex prooxidant ions.

The authors have never evaluated the properties of models in monomeric form.

Electrochemical data (cyclic voltammetry for instance) would undoubtedly support the measured data as they directly correlate to the redox properties.

Figures 1, 2 and 3 are of no interest in the manuscript and can be moved to the supporting information, with Table 1 summarising all the data. There is no table for the MPO tests (classical and SIEFED).

The discussion section is rather weak and the data rather sparse as the efforts were mainly focused on antioxidant activities. The article would gain in quality with a docking approach as suggested by the authors in the conclusion. Similarly, an approach to NET formation, as also suggested by the authors, would add to the quality.

Attention should be paid to the use of abbreviations that are not always consistent (e.g. PMN not described or EA not always used for ellagic acid).

The chemical structures must absolutely be incorporated into the text. Why use such complex references? Keeping the numbering used in the already accepted article would gain clarity.

The English language needs to be significantly improved. Some sentences do not make any sense. It is essential to proofread well and improve the quality of the English language. Compared to the previous article published by these authors in RSC Medicinal Chemistry, this article is disappointing in terms of writing quality and rigour.

Reviewer 2 Report

The authors tested antioxidant and effects on ellaigic acid and its analogues. As the solubility of ellaigic acid is low the authors used its synthetized analogs ( and tested their antioxidant activity using several adequate in vitro models incuding DPPH, ABTS, HRP and MPO activities tests and ROS production by neutrophiles.

The obtained results are scientifically important, but the manuscript should be carefully modified. In my opinion the obtained results and importance of the research is not very well described and there is a lot of information that are not relevant to their research in the manuscript. Although, the changes of redox status are an important factor that is characteristic for malaria, the authors actually do not provide no data about effects of ellaigic acid and its analogs on malaria. They should add some comments about possible effects on malaria treatment, but the authors should focus on obtained data. From this reason, starting from the title the manuscript should be modified not focusing and discuss about possible effects on malaria, but writing about  structure antioxidant effects relationships, maybe to add some literature data about solubility of the ellaigic acid analogs and to discus differences between tested analogs, to compare their effects, to recommend what analogs are the ones with the highest activity, which ones have the highest potential to be used in treatment of malaria but in my opinion the other conditions that are associated with changes in redox status, too.

The abbreviation introduction and use in text should be carefully cheked keeping in mind that the abbreviation should be introduced the first time when the term is mention in text and if the abbreviation is introduced it should be constantly used in text.

There is also some problem with symbols throughout the manuscript and it should be corrected before submitting new improved version of the manuscript. 

Round 2

Reviewer 1 Report

The authors have significantly improved the quality of their publication and taken into account most of the comments raised by this reviewer. As far as I am concerned, the article can now be accepted for publication in Molecules.

Author Response

Dear Editor,

On behalf of all the authors, I would like first to thank you for giving us the opportunity to submit the manuscript revised version.  My special thanks also go to the reviewer who made interesting suggestions to improve the final version of the manuscript.

Even though the reviewer 1 did not ask any change for the second round of the examination, the revised manuscript is attached in order to show all the changed made on the final manuscript.

I sincerely hope that everything is fine now.

Mouithys-Mickalad

Reviewer 2 Report

The present form of the manuscript is improved compared to previous version.

Still, I suggest to the authors to change the title and the structure of the introduction section. Malaria could be mention but it is not in the focus of this study, but some antioxidant properties of ellaigic acid derivates are tested. Perhaps, the authors could start introduction with paragraph about ellaigic acid and than mention effects on malaria,

The presentation of results should be further improved, too. The authors have figures 4, 5 and 6, so the numbering of the figures should be corrected. At the same time, the figure 6 should be black and white to be in ssame format as figures 4 and 5. 

Author Response

Dear Editor,

On behalf of all the authors, I would like first to thank you for giving us the opportunity to submit the manuscript revised version.  My special thanks also go to the reviewer who made interesting suggestions to improve the final version of the manuscript.

All the concerns raised by the reviewer have been corrected. The title and introduction have been modified accordingly. The numbers of figures (1, 2, and 3) are now fine and the two last graphiques also changed. The graphical abstract has been added at the end of the revised manuscript.

I sincerely hope that everything is fine now.